# SOCS1 Inhibits IL-6-Induced CD155 Overexpression in Lung Adenocarcinoma

**DOI:** 10.3390/ijms252212141

**Published:** 2024-11-12

**Authors:** Mario Marroquin-Muciño, Jesus J. Benito-Lopez, Mario Perez-Medina, Dolores Aguilar-Cazares, Miriam Galicia-Velasco, Rodolfo Chavez-Dominguez, Sergio E. Meza-Toledo, Manuel Meneses-Flores, Angel Camarena, Jose S. Lopez-Gonzalez

**Affiliations:** 1Laboratorio de Cancer Pulmonar, Departamento de Enfermedades Cronico-Degenerativas, Instituto Nacional de Enfermedades Respiratorias “Ismael Cosio Villegas”, Mexico City 14080, Mexico; mario.mm@ciencias.unam.mx (M.M.-M.); mperezm1518@alumno.ipn.mx (M.P.-M.); daguilarc@iner.gob.mx (D.A.-C.); miriam.galicia@iner.gob.mx (M.G.-V.); rodolfo_chvz@comunidad.unam.mx (R.C.-D.); manuelmeneses707@gmail.com (M.M.-F.); 2Laboratorio de Quimioterapia Experimental, Departamento de Bioquimica, Escuela Nacional de Ciencias Biologicas, Instituto Politecnico Nacional, Mexico City 11340, Mexico; smezat@ipn.mx; 3Posgrado en Ciencias Biologicas, Universidad Nacional Autonoma de Mexico, Mexico City 04510, Mexico; 4Departamento de Patologia, Instituto Nacional de Enfermedades Respiratorias “Ismael Cosio Villegas”, Mexico City 14080, Mexico; 5Laboratorio de Inmunobiologia y Genetica, Instituto Nacional de Enfermedades Respiratorias “Ismael Cosio Villegas”, Mexico City 14080, Mexico; ang_edco@yahoo.com.mx

**Keywords:** SOCS1, IL-6 CD155, PVR, lung adenocarcinoma, immunotherapy

## Abstract

CD155, also known as the poliovirus receptor (PVR), is a crucial molecule in the development and progression of cancer, as its overexpression favors immune evasion and resistance to immunotherapy. However, little is known about the mechanisms that regulate its overexpression. Proinflammatory factors produced by various cellular components of the tumor microenvironment (TME) have been associated with CD155 expression. We analyzed the effect of interleukin (IL)-6 on CD155 expression in lung adenocarcinoma. We found a positive relationship between mRNA and protein levels. This correlation was also observed in bioinformatics analysis and in biopsies and serum from patients with lung adenocarcinoma. Interestingly, lung adenocarcinoma cell lines expressing suppressor of cytokine signaling 1 (SOCS1) did not show increased CD155 levels upon IL-6 stimulation, and SOCS1 silencing reverted this effect. IL-6 and SOCS1 are critical regulators of CD155 expression in lung adenocarcinoma. Further basic and clinical studies are needed to define the role of these molecules during tumor development and to improve their clinical impact as biomarkers and targets for predicting the efficacy of immunotherapies. This study deepens the understanding of the intricate regulation of the immune checkpoints mediated by soluble factors and allows us to devise new ways to combine conventional treatments with the most innovative anticancer options.

## 1. Introduction

Lung cancer is one of the most prevalent cancers and a leading cause of death worldwide [1]. Lung cancer is classified into two major groups, small cell lung cancer (SCLC) and non-small cell lung cancer (NSCLC), which account for approximately 80–85% of lung cancer cases [2]. Lung adenocarcinoma is the most common histological type of NSCLC, representing 40–45% of cases. Lung adenocarcinoma is classified into five main histological subtypes that influence patient prognosis: lepidic, acinar, papillary, micropapillary, and solid [3]. Understanding the cellular processes and molecular alterations involved in the development and progression of lung cancer has led to significant advances in diagnostic and therapeutic strategies. Consequently, conventional and targeted therapies and immune checkpoint inhibitors (ICI) have been approved as options for cancer treatment. Unfortunately, not all patients exhibit a durable clinical response to these treatments [4,5,6].

In addition to the genetic and epigenetic alterations detected in tumor cells, particular focus has been given to the tumor microenvironment (TME). In particular, the immune system plays a vital role in preventing tumor progression by eliminating tumor cells. However, tumor cells exhibit several mechanisms that dampen the immune response to escape detection and elimination. One of these mechanisms is the overexpression of membrane proteins known as immune checkpoints (ICs), which provide inhibitory signals to the immune cells. Multiple immune checkpoints and their ligands, such as PD-1/PD-L1, CTLA-4/CD80/CD86, TIM-3/Galectin-9, LAG-3/MHC, and TIGIT/CD155, have been reported to date. Overexpression of these molecules in immune and tumor cells has been associated with poor prognosis in cancer patients [7]. Immunotherapy using monoclonal antibodies against diverse immune checkpoints called immune checkpoint inhibitors (ICI) seeks to reinvigorate the anti-tumor functions of the immune system [8]. ICI, alone or in combination with chemotherapy, has been shown to improve the overall survival of patients compared with conventional therapies [9]. The success of ICI depends on several host-intrinsic and tumor-intrinsic factors, such as the expression level, production of soluble isoform(s), or coexpression of multiple ICs [10,11]. Some reports have shown that pro-inflammatory factors can regulate the overexpression of immune checkpoints. For instance, PD-L1 expression is induced in tumor cells by IFN-γ/STAT-1 [12] and IL-6/STAT3 signaling [13,14]. Therefore, the study of TME factors that regulate IC expression is of great importance in the development of biomarkers or potential therapeutic targets to improve ICI success.

CD155, also known as poliovirus receptor (PVR), binds to TIGIT, CD226, and CD96 on immune cells to modulate the immune response in a similar way to the CD28/CTLA-4:CD80/CD86 network [15]. In addition, CD155 promotes proliferation, migration, and metastasis [16], and its overexpression has been associated with poor prognosis in cancer patients [17]. However, the factors that induce CD155 expression in the TME have been poorly described. CD155 expression is induced by sonic hedgehogs during embryonic development [18], TLR ligands in dendritic cells [19], and DNA damage following chemotherapeutic treatment [20]. Solecki et al. [21] reported that the human promoter of CD155 harbors a binding sequence for the transcription factor NF-κB, suggesting that pro-inflammatory signals may induce its expression. Recently, it has been reported that in multiple myeloma and hepatocellular carcinoma, CD155 expression is induced by IL-8, activating NF-κB signaling, and IL-22 activates STAT3, respectively [22,23].

Therefore, other pro-inflammatory factors may regulate CD155 expression. IL-6 is a pro-inflammatory cytokine produced in the TME by stromal, immune, and tumor cells and has been associated with tumor pathogenesis, progression, and treatment resistance in lung adenocarcinoma [24,25,26]. Classic and trans-signaling pathways mediate IL-6 signaling. In both cases, the binding of IL-6 to its receptor allows for its interaction with glycoprotein gp130 and the subsequent activation of JAK kinases and phosphorylation of the transcription factor STAT3 [27].

This study analyzed the correlation between IL-6 and CD155 in a lung adenocarcinoma cohort from The Cancer Genome Atlas (TCGA). Since we found a positive correlation between the expression of these molecules, we analyzed the effect of IL-6 stimulation on CD155 expression in the four lung adenocarcinoma cell lines. The correlation between these molecules was further studied in lung adenocarcinoma biopsies and patient serum samples. We found that IL-6 induced CD155 expression in the studied cell lines. Interestingly, the lung adenocarcinoma cell lines that displayed low expression of the negative regulator Suppressor of Cytokine Signaling 1 (SOCS1) showed increased CD155 expression after IL-6 stimulation, in contrast to those that showed high SOCS1 expression.

Furthermore, we found that this inverse relationship between SOCS1 and CD155 was maintained in the TCGA lung adenocarcinoma cohort. To further investigate the role of SOCS1 in IL-6-induced CD155 expression, we performed DsiRNA-mediated knockdown and confirmed that SOCS1 silencing restored the effect of IL-6 stimulation on CD155. Our results suggest that SOCS1 could be of particular relevance as a prognostic biomarker for the success of immunotherapies because it regulates the overexpression of CD155 in lung adenocarcinoma.

## 2. Results

### 2.1. Expression of IL-6 and CD155 in Lung Adenocarcinoma

The expression of IL-6 and CD155 at the transcript level was evaluated in patients from the TCGA lung adenocarcinoma cohort (*n* = 505). The data did not show significant changes in IL-6 expression among lung adenocarcinoma stages (*p* = 0.19) (Figure 1A, top). In contrast, CD155 expression was higher in advanced lung adenocarcinoma (Figure 1A, bottom). Regarding the relationship between IL-6 and CD155 (Figure 1B), we observed a weak but positive correlation (R = 0.25; *p* = 1.6 × 10^−8^). Interestingly, when categorizing lung adenocarcinoma patients according to clinical stage (Figure 1C), the correlation was higher in stage III patients (R = 0.45, *p* = 1.05 × 10^−45^) than in early stages. This suggests that the impact of IL-6 on CD155 expression may be more relevant in the advanced stages of lung adenocarcinoma, in which most patients are diagnosed.

### 2.2. IL-6-Induced CD155 Expression in Lung Adenocarcinoma Cell Lines

To analyze the effect of IL-6 on CD155 expression, four lung adenocarcinoma cell lines were exposed to low (10 ng/mL) and high (50 ng/mL) concentrations of IL-6 in serum-free medium. After stimulation, the mRNA and whole protein extracts were obtained. Figure 2A shows that IL-6 induced CD155 mRNA expression in all the four cell lines tested. However, Western blotting showed that the NCI-H1437 cell line, isolated from a patient with stage I lung adenocarcinoma and the A-549 cell line augmented CD155 levels upon IL-6 exposure. Interestingly, in 3B1A and NCI-H1573 cell lines derived from patients with advanced stages of lung adenocarcinoma, basal levels of CD155 protein were not affected by IL-6 stimulation (Figure 2B). We noticed that the pattern and intensity of the CD155 bands differed between the cell lines analyzed. In cell lines 3B1A and NCI-H1573 from advanced stages, two bands were detected, whereas in cell line A-549, only one low molecular weight band was observed, and cell line NCI-H1437 showed only the upper band (Figure 2B). To analyze whether the cell lines produced soluble isoforms of CD155 that could be related to the previous results, supernatants were collected and analyzed by ELISA.

All the cell lines released soluble CD155 (Figure 2C). The A-549 cell line produced approximately twice as much soluble CD155 as the other cell lines tested, and IL-6 exposure at 50 ng/mL increased 0.5-fold the CD155 release. In contrast, cell lines in advanced tumor stages showed slightly reduced soluble CD155 production after IL-6 stimulation. Interestingly, the NCI-H1437 cell line did not show modified release of this molecule (Figure 2C). This may be explained by the production of the membrane isoform in this cell line, as suggested by Western blot results (Figure 2B). To verify that IL-6 stimulation increases CD155 and soluble and membrane isoforms, we employed the JAK2 inhibitor AG-490, which showed that the addition of this inhibitor reduced IL-6-induced overexpression of CD155 in the A-549 and NCI-1437 cell lines (Figure 2D).

### 2.3. SOCS1 Negatively Regulates CD155 Expression in Lung Adenocarcinoma Cell Lines

The SOCS family of proteins plays an important role in inflammation by inhibiting cytokine signaling through the JAK-STAT signaling pathway. Studies have demonstrated that the SOCS family proteins regulate inflammatory pathways, including IL-6. Moreover, some studies have described the role of SOCS1 in the development and progression of cancer [28,29,30]. Because proinflammatory factors induce CD155 expression, we hypothesized that high SOCS1 expression negatively regulates CD155 expression. Data from the lung adenocarcinoma cohort from TCGA showed that patients with high SOCS1 expression showed low CD155 expression, and patients with low SOCS1 expression displayed high CD155 expression (Figure 3A), suggesting that SOCS1 may negatively regulate CD155 expression. Thus, we evaluated SOCS1 expression in lung adenocarcinoma cell lines. Notably, the 3B1A and NCI-H1573 cell lines derived from advanced stages of lung cancer showed high levels of SOCS1, whereas the A-549 and NCI-H1437 cell lines expressed barely detectable or minimal levels of SOCS1 (Figure 3B). This may be related to the observed differences in CD155 expression after IL-6 stimulation. To explore this hypothesis, we performed DsiRNA-mediated SOCS1 knockdown in 3B1A and NCI-H1573 cell lines and evaluated the effect of IL-6 stimulation on CD155 expression. By Western blotting (Figure 3C) and immunofluorescence (IF) (Figure 3D), we found that the knockdown of SOCS1 leads to an increase in CD155 expression upon IL-6 stimulation.

### 2.4. Correlation Between IL-6 and CD155 in Lung Adenocarcinoma Patients

In the tumor microenvironment (TME), a complex network of factors could be involved in CD155 overexpression; therefore, we evaluated whether the correlation between IL-6 and CD155 is preserved in lung adenocarcinoma tissues. We performed immunohistochemistry (IHC) staining for IL-6 and CD155 in the biopsies of patients with primary lung adenocarcinoma. We identified cases with high and low staining representing different expression levels of these molecules (Figure 4A). Regarding CD155, some cases showed membrane, cytoplasmic, or membrane and cytoplasmic staining, which may be related to the differential expression of the CD155 isoforms (Figure 4A). When comparing the relative expression of CD155 among histologic subtypes of lung adenocarcinoma, we found that solid-type tumors showed the highest CD155 expression compared to other less aggressive lung subtypes (Appendix A), which is in accordance with previous reports [31,32].

The positive correlation found in tumor biopsies (Figure 4B) suggests that although several factors may be involved in CD155 regulation, the elevated level of IL-6 in the TME plays an important role in the induction of CD155 overexpression in tumor cells. Moreover, high CD155 expression in tumor cells was related to lower CD8+ T cell infiltration (Figure 4C), similar to that previously reported in hepatocellular carcinoma [33]. Finally, we evaluated whether a correlation between CD155 and IL-6 was detected in the serum of the patients. We found a positive correlation, suggesting that systemic levels may be used as predictive biomarkers of this phenomenon (Figure 4D).

## 3. Discussion

CD155 has been identified as a critical molecule for cancer development and pathogenesis. CD155 has minimal expression in healthy tissues but is overexpressed in tumor cells. CD155 interacts with T cell immunoglobulin and the ITIM domain (TIGIT) to deliver an inhibitory signal that dampens lymphocyte activation. Moreover, high CD155 expression in tumor cells and patient serum has been reported to favor resistance to conventional therapies and immunotherapy [23,34,35]. Some reports have shown that proinflammatory cytokines induce CD155 overexpression in tumor cells. Bone marrow stroma-derived IL-8 and T lymphocyte-derived IL-22 induce CD155 expression in multiple myeloma [22] and hepatocarcinoma [23], respectively. The human CD155 promoter has been reported to have binding motifs for transcription factors such as NF-κB and STAT3.

Interleukin-6 is a pro-inflammatory cytokine produced in the tumor microenvironment by immune, stromal, and tumor cells. IL-6 has paradoxical effects, sometimes showing antitumor or protumor effects [36,37]. In the latter scenario, IL-6 suppresses antitumor immunity and promotes tumor progression and resistance to chemotherapy [38]. Most tumor cells overexpress IL-6, and activation of the IL-6/STAT3 signaling pathway promotes tumorigenesis by regulating multiple signaling pathways implicated in the hallmarks of cancer [39]. High serum levels of IL-6 have been associated with poor prognosis and high mortality, whereas patients with reduced levels of IL-6 respond better to conventional therapies [40,41,42]. Recently, Huseni et al. [43] reported that IL-6 levels alter STAT3-mediated BATF induction in cytotoxic T lymphocytes (CTLs), leading to their dedifferentiation, which negatively affects PD-L1 treatment and elicits immunotherapy resistance.

In this study, we explored whether IL-6 regulates CD155 expression in lung adenocarcinoma. The positive correlation found in the lung adenocarcinoma data from TCGA between the expression of IL-6 and CD155 suggests an interplay between the expression of these molecules. Our results demonstrate that IL-6 induces CD155 expression in lung adenocarcinoma cells. Based on our results, this cytokine is responsible, at least in part, for stimulating the transcription and production of CD155. This phenomenon may be particularly relevant in the early stages, represented by the A-549 and NCI-H1437 cell lines, compared with the advanced stages, represented by the 3B1A and NCI-H1573 cell lines. The heterogeneity observed in cell lines with respect to the level of IL-6 expression and cell localization of CD155 was also observed in lung adenocarcinoma tissues. In addition to membrane CD155, soluble isoforms can alter interactions between IC and its ligands, leading to resistance to immunotherapy; therefore, evaluating the production of different CD155 isoforms by tumor cells will be relevant for the success of immunotherapies.

SOCS molecules are negative regulators of the cytokine-induced Janus Kinase-Signal transducer and activator of the transcription (JAK-STAT) signaling pathway. Starr et al. [29] and Nicholson et al. [44] demonstrated that IL-6 induces rapid expression of SOCS1. Recently, it has been reported that this molecule and even the SOCS1-mimetic peptide Tkip [45,46] can modulate anti-tumor immunity depending on its expression level in anti-tumor lymphocytes and tumor cells via diverse mechanisms [47]. SOCS1 has been reported to inhibit proinflammatory pathways by hindering the recruitment and phosphorylation of STAT proteins by interacting with p53 and ATM in a transcriptional complex and by proteasomal degradation of targets such as p65 [48].

Interestingly, it has been reported that the expression of the regulator SOCS1 prevents the progression of various types of tumors through various biological activities [49]. In addition, several experimental models of inflammatory and autoimmune diseases have highlighted the importance of SOCS1 in regulating T cell activation, acting as a “non-classical” checkpoint and a tumor suppressor [50]. IL-6 and several other cytokines upregulate SOCS1 expression, which in turn regulates intracellular signaling induced by IL-2, IL-6, IL-13, and IFN-γ [51].

In this study, we found that 3B1A and NCI-H1573 cell lines, representative of advanced stages of lung adenocarcinoma, constitutively express SOCS1. Silencing of SOCS1 allowed IL-6 to induce CD155 overexpression; thus, constitutive expression of SOCS1 in tumor cells prevented CD155 overexpression upon IL-6 stimulation and may account for the weak correlation found in tumors. Finally, although several TME factors could regulate CD155 expression in tumor cells, the association between CD155 and IL-6 observed in cell lines was maintained in lung adenocarcinoma tissues and patient serum.

Therefore, heterogeneity among tumor cells influences the response to IL-6 stimulation. On the one hand, tumor cells with low or null SOCS1 expression, such as the A-549 and NCI-H1437 cell lines, would overexpress CD155 in response to IL-6. In contrast, elevated SOCS1 expression prevented CD155 expression, even in the presence of high IL-6 levels. Thus, SOCS1 expression may play a central role in the success of various therapeutic strategies. In the case of anti-TIGIT antibodies currently reported in preclinical and clinical trials [52], and particularly to Tiragolumab treatment, low SOCS1 expression would reflect high CD155 expression and may predict a better response therapeutic strategy in combination with anti-PD-L1 (CYTYSCAPE, NCT03563716) [53].

In contrast, high SOCS1 expression reflects low CD155 expression, which has been reported to impair the success of the PD-1/PD-L1 blockade. Moreover, we previously reported that drug-tolerant/persister (DTP) cells upregulate SOCS1 to survive cisplatin exposure [54]. Further studies are needed to evaluate the effect of SOCS1 expression on combining conventional therapies with immunotherapy or immunotherapy combinations, particularly in lung cancer [55,56].

In conclusion, this study showed that IL-6 induces CD155 expression in tumor cells and identified SOCS1 as a negative regulator of this process. The regulation of CD155 by IL-6 and SOCS1 and its effect on immunotherapies and combined therapies requires further studies.

## 4. Materials and Methods

### 4.1. Dataset Preparation

To investigate the relationship between IL-6 and CD155, we performed a correlation analysis between the expression levels of IL-6 and CD155 using RNA-seq data from the lung adenocarcinoma cohort of the Cancer Genome Atlas (TCGA). RNA-seq data and clinical information were retrieved using R (v 4.1.1) package RTCGA (v 1.22.0). Data were curated to select only primary tumor samples (*n* = 505) and eliminate duplicated samples. Raw counts were normalized using relative expression (transcripts per million/TPM), and the normal distribution of IL-6 and CD155 counts was tested. Spearman’s test was used to analyze the correlation between IL-6 and CD155 expression in the cohort of patients selected and by clinical stage. Statistical significance was set at *p* < 0.05.

### 4.2. Lung Adenocarcinoma Cell Lines

Lung adenocarcinoma cell lines A-549 (CVCL_0023) and NCI-H1437 (CVCL_1472), isolated from a patient with stage I lung adenocarcinoma, and NCI-H1573 (CVCL_1478), isolated from a patient with stage IV lung adenocarcinoma, were acquired from the ATCC (Manassas, VA, USA). We also included a 3B1A cell line previously obtained in our laboratory [57]. All cell lines were derived from pleural effusions of untreated patients with lung adenocarcinoma.

### 4.3. Cell Culture and IL-6 Stimulation

Cell lines were cultured in a humidified atmosphere incubator with 5% CO_2_ at 37 °C in RPMI-1640 medium containing 10% fetal bovine serum (FBS) and 1% antibiotics (complete medium). A-549 cells were seeded at 7.5 × 10^4^ cells/mL, and NCI-H1437, NCI-H1573, and 3B1A at 2 × 10^5^ cells/mL. After allowing the cells to adhere to plastic for 24 h, the complete medium was discarded, and serum-free medium was added. After 6 h of serum starvation, the cells were exposed to media (control), 10 ng/mL, or 50 ng/mL of human recombinant IL-6 (Cat. No. 206-IL, R&D, Minneapolis, MN, USA). During stimulation, routine microscopic monitoring was performed to observe any possible changes in cell morphology. In addition, to confirm that the observed effect was IL-6 dependent, the JAK2 inhibitor, Tyrphostin AG490 (Cat. No. S1143; Selleckchem, Houston, TX, USA), was added at a concentration of 15 μM prior to IL-6 addition.

### 4.4. Quantitative RT-PCR Analysis

After stimulation for 6 h, the exposed cells were trypsinized. Total RNA was extracted using the PureLink RNA Mini Kit (Cat. No. 12183018A, Thermo Scientific, Waltham, MA, USA) according to the manufacturer’s instructions. The cDNA was generated using a high-capacity cDNA reverse transcription kit (Cat. No. 4368814; Applied Biosystems, Carlsbad, CA, USA, Thermo Scientific) with a Master Cycle EP gradient S (Eppendorf, Framingham, MA, USA). Quantification and evaluation of cDNA were performed using Nanodrop 2000 (Thermo Fisher Scientific) by detecting a 260/280 absorbance ratio higher than 2.0. qPCR was performed using the Step One Plus Real-time PCR System (Applied Biosystems, Thermo Fisher Scientific) with the TaqMan Gene Expression probe for CD155 (Hs00197846_m1). Data were analyzed with respect to the expression of GAPDH (Hs99999905_m1), and the relative expression was calculated using the 2^−ΔΔCt^ method. Three independent experiments were performed.

### 4.5. SDS-PAGE and Western Blot Analysis

Cells were lysed after 18 h of IL-6 exposure using 2% Triton X-100 solution containing a Halt Protease Inhibitor Cocktail (Cat. No. 78438, Thermo Fisher Scientific). Proteins were quantified using the MicroBCA Protein Assay Kit (Cat. No. 23235, Thermo Fisher Scientific). Cell lysates (25 μg) were resolved using sodium dodecyl sulfate-polyacrylamide gel electrophoresis and transferred to a nitrocellulose membrane. After blocking non-specific sites with 2% BSA in PBS for 30 min, the membranes were incubated at 4 °C overnight with the following primary antibodies: CD155 (Cat. No. ab267788; 1:500 dilution, Abcam, Waltham, MA, USA) or SOCS-1 (Cat. No. GTX100657, 1:300 dilution, Genetex, Irvine, CA, USA). After washing, membranes were incubated with biotinylated anti-rabbit secondary antibodies (Cat. No. 65-6140, dilution 1:1000, Invitrogen, Waltham, MA, USA) for 1 h at RT. After washing, the membranes were incubated with ABC complex/HRP (PK-6100, dilution 1:400, Vector, Newmarket, AK, USA) at room temperature (RT) for 30 min. For re-probing, the membranes were incubated with Tween/SDS/glycine stripping buffer pH 2.2 at 37 °C for 20 min and incubated in blocking solution at RT for 30 min before adding the anti-β-actin antibody (Cat. No. A-1978, 1:10,000 dilution, Sigma, Burlington, MA, USA). Biotinylated anti-mouse IgG (Cat. No. 31803, dilution 1:1000, Invitrogen) antibody was used following the above procedure. Protein bands were visualized employing the ECL Select Western Blotting Detection Reagent (Cat. No. RPN2235, Amersham Biosciences, Buckinghamshire, UK). Bands were detected in the ChemiDoc Image System (Bio-Rad Laboratories, Hercules, CA, USA). Three independent experiments were performed in duplicates.

### 4.6. Quantification of CD155 in Supernatants of Culture Cell Lines by ELISA

Supernatants from cells cultured in T-80 flasks and exposed for 18 h to IL-6 were collected. Soluble CD155 was quantified using the Human CD155/PVR/Poliovirus Receptor ELISA Kit, PicoKine (Cat. No. EK1831, Boster Biological Technology, Pleasanton, CA, USA). The assay was performed according to the manufacturer’s instructions. Optical density was measured at 450 nm using a Multiskan Ascent spectrophotometer (Thermo Fisher Scientific). Replicates from two independent experiments were performed.

### 4.7. SOCS1 Knockdown

The cells were seeded as described above. After 24 h, cells were washed with FBS-free medium and maintained under serum starvation for 4 h. Knockdown was performed employing the TriFECTa RNAi Kit in OptiMEM medium (Cat. No. 31985, Thermo Fisher, Waltham, MA, USA) according to the manufacturer’s instructions. The cells were incubated with 3 μL of lipofectamine 3000 (Cat. No. L3000-015 Invitrogen, USA) and mixes of DsiRNAs SOCS1 (HS.Ri.SOCS1.13.1-3, IDT, Arlington, VA, USA) at a final concentration of 10 nM for 48 h, at which the maximum knockdown was obtained. Cell cultures containing Lipofectamine were used as negative controls (mock). The DsiRNA sequences used for knockdown are provided in Appendix A. Two independent experiments were performed.

### 4.8. Immunofluorescence for CD155 Induced by IL-6

Cell lines were seeded in 4-well chamber slides (Lab-TeK, Thermo Fischer Scientific), serum-starved, and exposed to IL-6, as described above. The cells were then fixed with 96% ethanol. After washing and blocking non-specific binding, the slides were incubated with the monoclonal antibody anti-CD155 (Cat. No. ab267788, 1:100 dilution). The next day, the slides were incubated with Alexa Fluor488 F(ab’)2 goat anti-rabbit antibody (Cat. No. A11070; dilution, 1:250; Invitrogen) for 4 h. After washing, DAPI (Cat. No. 62248, dilution 1:150, Sigma-Aldrich, St. Louis, MO, USA) was added for nuclear staining, and Vectashield was added for mounting (Cat. No. H1000 Vector Laboratories, Burlingame, CA, USA). Two independent experiments were performed.

### 4.9. Biopsy Collection

Biopsies from 39 untreated patients with primary lung adenocarcinoma were obtained from the archive of the Pathology Department of the Instituto Nacional de Enfermedades Respiratorias Ismael Cosio Villegas. The demographic and clinical characteristics of the study cohort are shown in Appendix A. From the biopsy, 3–4-μm serial tissue sections were obtained. In each case, one slide was stained with hematoxylin and eosin (H&E) to confirm the histological criteria for lung adenocarcinoma. The present study followed the principles of the Declaration of Helsinki and the ethical guidelines of our institution and was approved by the Scientific, Bioethics, and Biosafety Committee (B03-22). Patient consent was waived for the use of tissue blocks from lung adenocarcinoma patients obtained from the repository of residual biological material from the archives of the Pathology Department of the institution.

### 4.10. Quantification of IL-6 and CD155 Expression in Tumor Cells and Infiltrating CD8+ T Lymphocytes in Lung Adenocarcinoma Biopsies

Immunostaining was performed as previously described [26]. Slides were deparaffinized at 60 °C for 20 min and rehydrated. All slides were subjected to heat-induced epitope retrieval, using 0.01 M citrate buffer, pH 6.0 at 110 °C, 6 psi for 20 min in a NxGen decloaking chamber (Biocare Medical, Pacheco, CA, USA). Slides were treated with 3% (*v*/*v*) H_2_O_2_ in methanol for 20 min to block endogenous peroxidase activity, washed with phosphate-buffered saline (PBS), and incubated with PBS containing 2% normal serum for 30 min to block non-specific binding. Slides were then incubated at 32 °C for 1 h and overnight at 4 °C in a humidified chamber with primary anti-human antibodies: anti-CD155 (Cat. No. ab267788, dilution 1:100, Abcam, Cambridge, UK) or anti-IL-6 (Cat. No. bs-4587R, dilution 1:50, BIOSS, Woburn, MA, USA). Next day, tissue sections were washed three times with 0.1% *v*/*v* PBS-Tween and PBS and incubated with the corresponding species-specific biotin-labeled secondary antibodies Goat anti-rabbit (Cat. No. 65-6140, dilution 1:1000, Invitrogen), Goat anti-mouse (Cat. No. 31803, dilution 1:1000, Invitrogen) in a humidified chamber at 32 °C for 1 h. After washing, the tissue sections were incubated with ABC complex/HRP (Cat. No. PK-6100, dilution 1:150, Vector) at 32 °C for 30 min. The color was developed using H_2_O_2_ as the substrate and diaminobenzidine (DAB) as the chromogen, generating a brown color. Slides were mounted using Entellan (Cat. No. HX43610761, Merck, Lebanon, NJ, USA). The relative expression of IL-6 and CD155 was measured as the mean optical density (OD) of DAB in at least five different fields.

To investigate CD8+ T cell infiltration, we selected tissues (*n* = 28) with considerable tumor infiltrating lymphocytes (TILs), as observed by H&E staining. IHC was performed using an anti-CD8 antibody (Cat. No. ab4055, dilution 1:50; Abcam). CD8-positive cells were counted as positive particles. Mean counts of at least five different fields are presented. Analyses were performed using the ImageJ software (v1.53).

### 4.11. Quantification of IL-6 and CD55 in Plasma from Lung Adenocarcinoma Patients

Plasma was collected from 31 lung adenocarcinoma patients, and IL-6 and CD155 levels were quantified. The ELISA kit for Human IL-6 (Cat. No. BMS213.2, Thermo Scientific) was used to detect IL-6, and for CD155, the kit used for measuring this molecule in cell culture supernatants was employed. Assays were performed according to the manufacturer’s instructions. Optical density was measured at 450 nm using a Multiskan Ascent spectrophotometer (Thermo Fisher Scientific). Samples were analyzed in triplicate.

### 4.12. Statistical Analysis

The Shapiro–Wilk test was used to test for normal data distribution. Normal data were expressed as mean ± standard deviation (SD). Unless otherwise stated, ANOVA analyzed data, and Tukey’s test was used for multiple comparisons. TCGA lung adenocarcinoma cohort data are presented as the median ± interquartile range. Non-parametric Kruskal–Wallis and Dunn’s multiple comparison tests were used to compare data among the clinical stages. Spearman’s test was employed to test the correlation between IL-6 and CD155 in lung adenocarcinoma data from the TCGA and relative expression in tissues. Bioinformatic data were analyzed using R v.4.1.1. Statistical analyses were performed using GraphPad Prism 8 software (GraphPad Software, La Jolla, CA, USA). Statistical significance was set at *p* < 0.05.

## Figures and Tables

**Figure 1 ijms-25-12141-f001:**
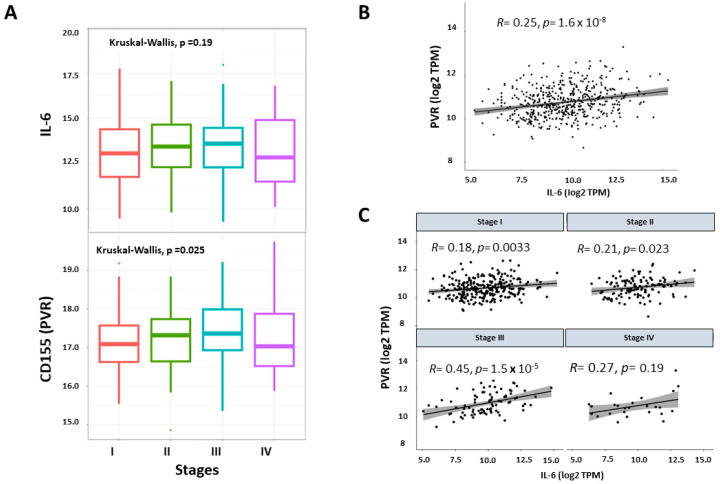
IL-6 and CD155 expression in lung adenocarcinoma data from TCGA. (**A**) IL-6 expression did not change among lung adenocarcinoma stages. CD155 (PVR) expression in stage III cases (*n* = 84) was higher than that in early stages I (*n* = 275) and II (*n* = 121). Patients with stage IV (*n* = 25). The minimum, 25th percentile, median, 75th percentile, and maximum values are shown. (**B**) Correlation between the expression of IL-6 and PVR (CD155) in the lung adenocarcinoma cohort (*n* = 505). (**C**) Correlation between IL-6 and CD155 among the clinical stages of the lung adenocarcinoma cohort.

**Figure 2 ijms-25-12141-f002:**
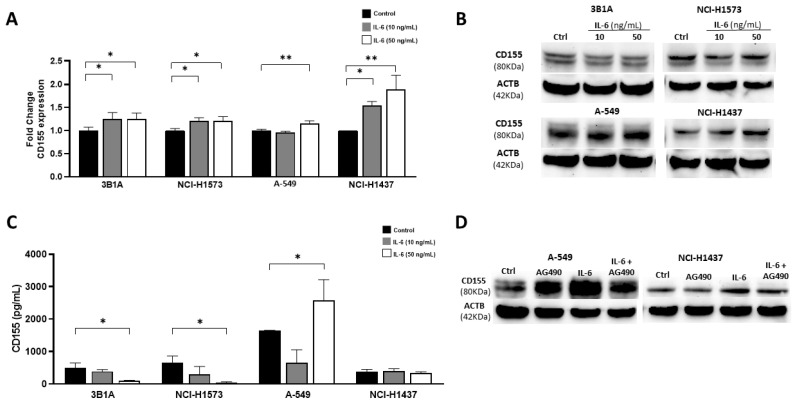
Effect of IL-6 stimulation on CD155 expression on lung adenocarcinoma cell lines. (**A**) IL-6 stimulation induced CD155 mRNA expression in the tested cell lines. (**B**) Upon IL-6 stimulation, CD155 protein expression increased in A-549 and NCI-H1437 cells, but no change was detected in the advanced stages 3B1A and NCI-H1573 cell lines. (**C**) Results show that CD155 concentration was augmented only in the supernatant of the IL-6-stimulated A-549 cell line. (**D**) Addition of the JAK2 inhibitor AG490 prevents IL-6-induced CD155 overexpression in the A-549 and NCI-H1437 cell lines. (**A**,**C**) are shown as mean + SD. * *p* < 0.05, ** *p* < 0.01.

**Figure 3 ijms-25-12141-f003:**
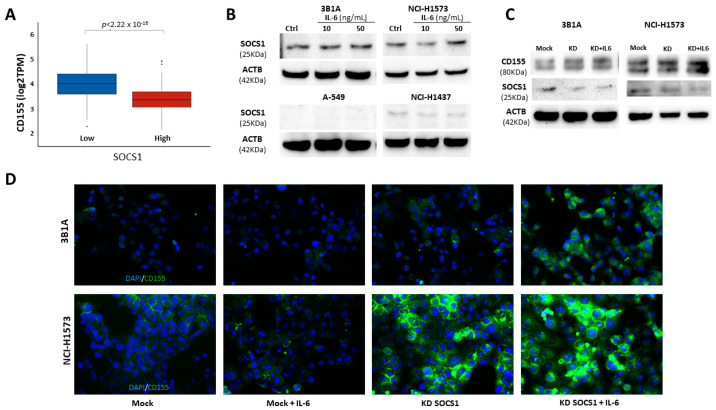
SOCS1 negatively regulates CD155 expression. (**A**) Comparison of CD155 expression between lung adenocarcinoma patients from TCGA with high and low SOCS1 expression. (**B**) Western blot analysis showed that the 3B1A and NCI-H1573 (advanced tumor stages) cell lines showed high SOCS1 expression compared to the A-549 and NCI-H1437 (tumor stage I) cell lines, and this expression was not significantly affected by IL-6 stimulation. (**C**) Western blotting and (**D**) immunofluorescence show that SOCS1 knockdown enables the 3B1A and NCI-H1573 cell lines to augment CD155 expression upon IL-6 stimulation. CD155 (green) and nuclear staining (blue). Magnification 400×.

**Figure 4 ijms-25-12141-f004:**
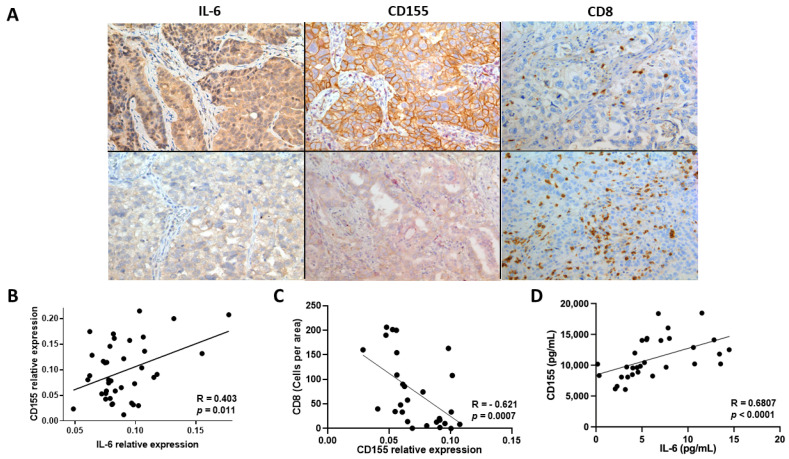
IL-6 and CD155 expression in patients with LUAD. (**A**) Representative micrographs of cases with high (top) and low (bottom) IL-6 and CD155 staining and cases with low (top) and high (bottom) CD8+ T cell infiltration in lung adenocarcinoma biopsies. IL-6 showed cytoplasmic staining, while CD155 was found in the membrane and cytoplasm of the tumor cells. Magnification 200×. (**B**) Positive correlation between the relative expression of IL-6 and CD155 in lung adenocarcinoma in tissues (*n* = 39). (**C**) Negative correlation between CD155 expression and CD8+ T-cell infiltration in lung adenocarcinoma tissues. (**D**) Correlation between serum IL-6 and CD155 concentrations in patients with primary lung adenocarcinoma (*n* = 31).

## Data Availability

The data generated in this study are available upon request from the corresponding authors.

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
