# Peer review of "SOCS1 Inhibits IL-6-Induced CD155 Overexpression in Lung Adenocarcinoma"

_ijms, 2024, doi:10.3390/ijms252212141_

Round 1
Reviewer 1 Report
Comments and Suggestions for Authors
This manuscript by Mario Marroquin Mucino, presented data on SOCS1 regulatory effect on IL-6 induced CD155 expression in lung adenocarcinoma. By approaching SOCS1 expressing lung adenocarcinoma cell line model, authors found that SOCS1 expression did not show any effect on CD155 upon IL-6 stimulation. However, SOCS1 silencing showed increased expression. Thus, the authors confirmed that SOCS1 suppress CD155 expression.
Below are my concerns which authors may consider addressing them;
1) Abstract-Good. Please describe PVR.
2) Introduction-Nicely written
3) Results.a) Its hard to follow different cell lines used. Particularly, in figure legends. please describe which cell line corresponds to which clinical stage, etc. b) Figure legends-please describe the statistical method used to calculate p value.
This manuscript is simple as it presented, easy to follow and understand.
Author Response
Thank you for your valuable comments. We have made the indicated amendments and believe that they have improved the current version.
- In lines 19 and 71, we have included the meaning of PVR (Poliovirus receptor) as an alternative name for CD155.
- To clarify information on the clinical stages of the cell lines used in our manuscript. We include this information throughout the text, particularly in the subsection IL-6-induced CD155 expression in lung adenocarcinoma cell lines and in the footnote to Figure 2. We hope that this will improve the comprehensibility of the manuscript. In addition, we highlight the statistical method used for each analysis in the Material and Methodology section.
Reviewer 2 Report
Comments and Suggestions for Authors
Overall, the manuscript is well prepared, here are some concerns on the analysis and report of the results:
1. First, the reported correlation between IL-6 and CD155 of 0.25 in TCGA (n = 505) and 0.2629 in the tumor samples (31) is weak. Each explain less that 10% of the other’s variation.
2. For cell lines exposed to low (10ng/mL) and high (50ng/mL) concentrations of IL-6 in serum free medium, how many samples were treated? It would be nice to have magnitudes of difference in CD155 expression between groups.
3. For figure 4 D, with the reported r of 0.2629, and 31 samples in analysis, the p-value should be around 0.15 (2-sided). The reported p-value there is not supported by the data.
Author Response
We are grateful for the comments made to our work, as they impact on a better understanding of the manuscript.
- We agree that the correlation value (R) obtained from TCGA data is weak. However, this value increases when analyzing data from stage III patients in this cohort. Although mathematically this relationship was weak, it is important to analyze the biological role of IL-6 present in the tumor microenvironment and whether it could influence CD155 expression.
- In this version, the number of experiments performed is included. You can find this information in the Materials and Methods section and the p value is indicated in the Fig. 2 and Supplementary Fig. 1.
- We apologize for the mistake. Considering your comment, we re-analyzed the data distribution and Shapiro-Wilkins test showed that data present non-normal distribution. Spearman’s test shows an R value of 0.6807 and p value <0.0001. This change is indicated in Figure 4D.